# Exploring socio-demographic determinants of breast self-examination practices among Jordanian women: insights from the 2023 population-based survey

Amr Ahmed Aly Ibrahim[1,2], Moaz Yasser Darwish[3], Sara Hosny El-Farargy[4,7], Nour Eldein Saad[5], Aya Khafage[6], Mahmoud Shaaban Abdelgalil[2,7] *

1 Faculty of Medicine, Menoufia University, Menoufia, Egypt, 2 Research Insights Arab Network, Cairo, Egypt, 3 Faculty of Medicine, Fayoum University, Fayoum, Egypt, 4 Faculty of Medicine, Benha University, Benha, Egypt, 5 Faculty of Medicine, Alexandria University, Alexandria, Egypt, 6 Faculty of Medicine, Menoufia University, Menoufia, Egypt, 7 Faculty of Medicine, Ain-shams University, Cairo, Egypt

* 29908068800596@med.asu.edu.eg

## Abstract

### Background

Breast cancer is the most prevalent cancer and the leading cause of cancer-related deaths among Jordanian women. Breast self-examination (BSE) plays a vital role in the early detection of breast cancer, improving survival rates. Despite its proven benefits, BSE remains underutilized in Jordan. This study aimed to explore the factors influencing BSE practices among married Jordanian women aged 20–49 years, utilizing data from the Jordan Population and Family Health Survey (JPFHS).

### Methods

The study analyzed data from the 2023 JPFHS, encompassing a representative sample of 12,595 Jordanian women aged 15–49. The study examined various socioeconomic, demographic, behavioral, and geographic variables. Socioeconomic and demographic factors included age, education level, wealth index, employment status, marital status, parity, and current pregnancy status. Behavioral factors encompassed smoking frequency and media consumption habits, such as internet use, the frequency of watching television, listening to the radio, and reading newspapers or magazines. Geographic variables included the type of residence (urban or rural) and the governorates where participants lived. Associations between these variables and BSE were assessed using multivariable logistic regression.

### Results

Among the 12,304 married women included in the analysis, 9,851 women reported not performing BSE, while 2,453 women indicated that they had. Multivariate analysis

**Data availability statement:** The data used in this study are third-party datasets from the Demographic and Health Surveys (DHS) Program, managed by ICF International. The authors do not have permission to publicly redistribute these data due to their sensitive, de-identified nature and DHS data access policies. Qualified researchers can access the data free of charge by creating an account and submitting a brief research proposal via the DHS Program website (https://dhsprogram. com). No special privileges were granted to the authors; all researchers follow the same application process. For data access inquiries, please contact the DHS Program at archive@ dhsprogram.com.

**Funding:** The author(s) received no specific funding for this work.

**Competing interests:** The authors have declared that no competing interests exist.

revealed that significantly better BSE practice was observed among older women (e.g., age 45–49 vs 20–24: AOR 3.08, p < 0.001), those with higher education levels (e.g., secondary vs no education: AOR 2.41, p = 0.027), and wealthier women (e.g., richest vs poorest: AOR 1.54, p = 0.023). Additionally, multiparous women, daily smokers, and women with frequent internet use and frequent reading of newspapers or magazines were also more likely to practice BSE. Regional differences showed that women in Ajloun, Aqaba, and Balqa were more likely to perform BSE, while women in Irbid and Mafraq had lower rates of BSE practice.

## Conclusion

To improve BSE rates among married women in Jordan, targeted health campaigns should focus on younger, less educated, and economically disadvantaged women, particularly in Irbid and Mafraq. Culturally sensitive education, digital platforms, and community outreach can raise awareness and address barriers like stigma and misconceptions, promoting proactive breast health practices nationwide. Future researchers are encouraged to further investigate cultural barriers toward BSE.

## Introduction

Breast cancer is a significant global health issue, with an estimated 2.3 million new cases diagnosed in 2020, resulting in approximately 685,000 deaths [1,2]. Projections suggest that these numbers could increase to 4.4 million cases by 2070 [2,3], In Jordan, a lower- to middle-income country with a rapidly growing population and a challenged economy, cancer is becoming an increasingly pressing health issue [4]. Currently ranked as the second leading cause of death, after cardiovascular diseases, cancer, particularly breast cancer, is a major concern [4].

In 2022, breast cancer was the most common cancer among females in Jordan and the leading cause of cancer-related mortality in females [5]. The incidence of breast cancer among Jordanian women has increased by 69% in 2015 compared to 2005. This increase is likely related to the screening and early diagnosis initiatives implemented by the Jordan Breast Cancer Program (JBCP) since its establishment in 2006 [6]. The overall 5-year survival rate was estimated at 64% among Jordanian women diagnosed with breast cancer between 1997 and 2002 [7]. In contrast, those diagnosed between 2011 and 2014, following the establishment of the JBCP, showed an improved 5-year overall survival rate of 84% [8].

Before the establishment of the JBCP, more than 70% of patients were diagnosed at an advanced stage of the disease [9]. Unfortunately, a significant proportion of breast cancer cases in Jordan are still diagnosed at advanced stages. Recent data reveals that 36% of patients are diagnosed at stage III, and 14.1% at stage IV, meaning over half of cases are detected at late stages [8]. This contributes to lower survival rates, with the overall 5-year survival rate for breast cancer patients in Jordan at 46% for those diagnosed at stages III and IV [8]. To address this, the Jordanian

national guidelines emphasize early diagnosis and treatment of breast cancer in symptomatic, average-risk patients, aiming to improve both quality of life and survival rates [10].

Screening methods like mammography, clinical breast examination, and breast self-examination (BSE) are crucial for the early diagnosis of breast cancer, which improves survival rates [10]. Due to its accessibility and affordability, BSE remains a vital tool for self-monitoring and early detection, even though mammography and clinical breast exams are the standard in clinical settings [11]. While the national Jordanian breast cancer screening and diagnosis guidelines acknowledges the controversy regarding BSE, it encouraged women to practice BSE between routine screening to facilitate early detection of breast changes [12].

However, in many places, including Jordan, BSE is not practiced despite its potential benefits [13–15]. BSE is a simple method for detecting early signs of breast cancer through visual inspection and palpation. It should be done 7–10 days after the menstrual period. For breastfeeding women, BSE should be performed after the breasts are emptied of milk. Pregnant or menopausal women are advised to choose a consistent day each month. A study by Al-Najar et al. revealed that while 90% of Jordanian women were aware of BSE, only half reported practicing it [13]. Similarly, data from the 2023 Jordan Population and Family Health Survey (JPFHS) indicates that 80.2% of women have never performed a BSE, underscoring a significant gap between awareness and practice [16].

Barriers to BSE in Jordan include cultural, structural, and informational challenges [17].Cultural norms, such as embarrassment, patriarchal family dynamics, and the need for male approval, often restrict women's autonomy in health-related decisions. Additionally, religious beliefs, stigma, fatalistic attitudes, and reliance on traditional medicine diminish the perceived importance of screening. Structural barriers include financial constraints related to the absence of health insurance and high screening costs as well as accessibility challenges, such as the distance to healthcare facilities. Informational barriers include limited education, fear of diagnosis, lack of motivation, and misconceptions about cancer further discourage BSE [17].

A previous study by Al-Rifai et al., using data from the 2012 Jordan Population and Family Health Survey (JPFHS) [18], identified several factors influencing breast cancer screening among women aged 20–49 years. The findings revealed that younger women, those living in rural areas or the southern region, those with minimal education, nulliparous women, and those with infrequent access to media were less likely to participate in breast cancer screening. However, the lack of updated research on the socioeconomic, behavioral, and reproductive factors associated with BSE limits our understanding of the current challenges. Also, the growing implementation of digital and televised awareness warrants further investigation.

This study aims to fill this gap by analyzing the socio-demographic correlates of BSE among adult ever-married women in Jordan, utilizing the most recent data from the 2023 JPFHS. The findings will provide valuable insights to guide targeted interventions and improve breast cancer prevention and early detection strategies in Jordan.

## Materials and methods

According to the guidelines of the Jordan Breast Cancer Program (JBCP) established in 2007, all women aged ≥20 years should perform a monthly BSE [10].

### Study design and data source

This study utilized data from the 2023 JPFHS, which included a nationally representative sample of 12,595 women aged 15–49 years from all 12 governorates of Jordan. A formal research proposal was submitted to the Demographic and Health Surveys (DHS) program website in order to access the dataset and gain approval to use the publicly available, de-identified, dataset for research purposes.

### Sampling methodology

Representativeness of the Jordanian populations was a major consideration for the JPFHS; therefore, they adopted a two-stage stratified cluster sampling approach. The first stage constituted of the selection of primary sampling units (PSUs)

based on population size and place of residence (urban or rural). In the second stage, households within each PSU were randomly selected. The Jordanian Department of Statistics implemented the survey between January and June 2023. The detailed sampling methodology is described in original JPFHS 2023 report [19].

### Dependent variable

The dependent variable for our study was breast self-examination practice, assessed through the question, "Have you performed breast self-examination in the past 12 months?" Responses were categorized as binary (Yes/No), consistent with methodologies adopted in previous studies utilizing DHS data [18,20].

### Inclusion criteria

The study included ever-married women aged 20–49 years who provided reliable responses of "yes" or "no" to the question regarding BSE practices.

### Exclusion criteria

Women with incomplete or missing data on BSE practices, as well as those who were unaware of breast self-examination or breast cancer, were excluded from the analysis. Additionally, women under the age of 20 were excluded, as BSE screening is not recommended for this age group [10].

### Independent variables

The independent variables in the study included age (categorized in 5-year groups: 20–24, 25–29, 30–34, 35–39, 40–44, and 45–49), highest educational level (no education, primary, secondary, and higher), wealth index (poorest, poorer, middle, richer, and richest), and employment status (currently working or not working). Body mass index (BMI) was categorized as underweight, normal weight, overweight, and obese according to CDC criteria [21], while parity was categorized as nulliparous, primiparous, and multiparous.

Current pregnancy status was classified as "no or unsure" and "yes." Marital status was classified into married, widowed, divorced, and no longer living together/separated. The analysis was limited to ever-married women because the JPFHS dataset did not include breast self-examination data for women who were never married, as shown in **Table 1**. Therefore, only ever-married women were included in the study.

Smoking frequency was categorized as "does not smoke," "every day," and "some days." Internet use in the past month was divided into "not at all," "less than once a week," "at least once a week," and "almost every day," while the frequency of watching television and listening to the radio were grouped into "not at all," "less than once a week," and "at least once a week." The frequency of reading newspapers or magazines was categorized as "not at all," "less than once a week," and "at least once a week." Additionally, the type of place of residence (urban or rural) and the governorates (Amman, Balqa, Zarqa, Madaba, Irbid, Mafraq, Jarash, Ajloun, Karak, Tafiela, Ma'an, and Aqaba) were included as variables. All these categorizations were pre-defined by the DHS and provided as such in the dataset, ensuring consistency with DHS coding schemes and comparability with other DHS-based studies

### Statistical analysis

Data were analyzed using SPSS version 24, with a weighted count applied according to DHS guidelines [22]. Weighted analysis was applied in accordance with DHS guidelines to ensure national representativeness, accounting for the complex sampling design, sampling probabilities, and non-response. The women's individual sample weight (v005), provided as an eight-digit variable with six implied decimal places, was divided by 1,000,000 for application in the analysis. Descriptive statistics were used to summarize the characteristics of the study population, presenting results as frequencies and

**Table 1. Sociodemographic characteristics by BSE status.**

| Variables | | Performed breast cancer self-examination | | | |
|---|---|---|---|---|---|
| | | No (N=9,851) | | Yes (N=2,453) | |
| | | Count | % | Count | % |
| Age in 5-year groups | 20-24 | 831 | 8.40% | 57 | 2.30% |
| | 25-29 | 1593 | 16.20% | 176 | 7.20% |
| | 30-34 | 1852 | 18.80% | 364 | 14.80% |
| | 35-39 | 1835 | 18.60% | 467 | 19.00% |
| | 40-44 | 1721 | 17.50% | 610 | 24.80% |
| | 45-49 | 2019 | 20.50% | 779 | 31.70% |
| BMI | Underweight | 93 | 1.70% | 9 | 0.70% |
| | normal weight | 1419 | 26.50% | 314 | 24.60% |
| | Overweight | 1854 | 34.60% | 407 | 31.90% |
| | Obese | 1986 | 37.10% | 546 | 42.80% |
| Current marital status | Never in union | 0 | 0.00% | 0 | 0.00% |
| | Married | 9235 | 92.10% | 2290 | 92.90% |
| | Living with partner | 0 | 0.00% | 0 | 0.00% |
| | Widowed | 280 | 2.80% | 72 | 2.90% |
| | Divorced | 499 | 5.00% | 98 | 4.00% |
| | No longer living together/separated | 8 | 0.10% | 4 | 0.20% |
| Childbirth | Nulliparity | 870 | 8.70% | 98 | 4.00% |
| | Primiparity | 1103 | 11.00% | 160 | 6.50% |
| | Multiparity | 8048 | 80.30% | 2206 | 89.50% |
| Currently pregnant | No or unsure | 9301 | 92.80% | 2376 | 96.40% |
| | Yes | 721 | 7.20% | 88 | 3.60% |
| Frequency smokes cigarettes | Does not smoke | 9272 | 92.50% | 2164 | 87.80% |
| | Every day | 578 | 5.80% | 243 | 9.80% |
| | Some days | 172 | 1.70% | 58 | 2.30% |
| Highest educational level | No education | 229 | 2.30% | 23 | 0.90% |
| | Primary | 677 | 6.80% | 89 | 3.60% |
| | Secondary | 5769 | 57.60% | 1353 | 54.90% |
| | Higher | 3347 | 33.40% | 999 | 40.50% |
| Respondent currently working | No | 8734 | 87.10% | 2051 | 83.20% |
| | Yes | 1288 | 12.90% | 413 | 16.80% |
| Wealth index combined | Poorest | 2128 | 21.20% | 296 | 12.00% |
| | Poorer | 2214 | 22.10% | 391 | 15.90% |
| | Middle | 2131 | 21.30% | 536 | 21.80% |
| | Richer | 1871 | 18.70% | 591 | 24.00% |
| | Richest | 1678 | 16.70% | 650 | 26.40% |
| Frequency of using the Internet last month | Not at all | 2284 | 22.80% | 367 | 14.90% |
| | Less than once a week | 109 | 1.10% | 30 | 1.20% |
| | At least once a week | 311 | 3.10% | 79 | 3.20% |
| | Almost every day | 7318 | 73.00% | 1987 | 80.70% |
| Frequency of watching television | Not at all | 1603 | 16.00% | 377 | 15.30% |
| | Less than once a week | 1801 | 18.00% | 484 | 19.60% |
| | At least once a week | 6619 | 66.00% | 1603 | 65.10% |
| | Almost every day | 0 | 0.00% | 0 | 0.00% |

*(Continued)*

**Table 1.** (Continued)

| Variables | | Performed breast cancer self-examination | | | |
|---|---|---|---|---|---|
| | | No (N=9,851) | | Yes (N=2,453) | |
| | | Count | % | Count | % |
| Frequency of listening to radio | Not at all | 7183 | 71.70% | 1625 | 66.00% |
| | Less than once a week | 1260 | 12.60% | 411 | 16.70% |
| | At least once a week | 1579 | 15.80% | 428 | 17.40% |
| | Almost every day | 0 | 0.00% | 0 | 0.00% |
| Frequency of reading newspapers or magazines | Not at all | 7747 | 77.30% | 1754 | 71.20% |
| | Less than once a week | 935 | 9.30% | 318 | 12.90% |
| | At least once a week | 1340 | 13.40% | 392 | 15.90% |
| | Almost every day | 0 | 0.00% | 0 | 0.00% |
| Type of place of residence | Urban | 9087 | 90.70% | 2291 | 93.00% |
| | Rural | 935 | 9.30% | 173 | 7.00% |
| Governorate | Amman | 4503 | 44.90% | 1209 | 49.10% |
| | Balqa | 530 | 5.30% | 151 | 6.10% |
| | Zarqa | 1287 | 12.80% | 374 | 15.20% |
| | Madaba | 172 | 1.70% | 48 | 2.00% |
| | Irbid | 2103 | 21.00% | 354 | 14.40% |
| | Mafraq | 456 | 4.60% | 48 | 1.90% |
| | Jarash | 252 | 2.50% | 55 | 2.20% |
| | Ajloun | 129 | 1.30% | 75 | 3.00% |
| | Karak | 233 | 2.30% | 50 | 2.00% |
| | Tafiela | 92 | 0.90% | 21 | 0.80% |
| | Ma'an | 125 | 1.20% | 26 | 1.10% |
| | Aqaba | 141 | 1.40% | 53 | 2.10% |

percentages. To investigate factors influencing BSE practices, multivariable logistic regression analyses were conducted, coding the dependent variable as 0 for respondents who answered "No" and 1 for those who answered "Yes," with the "No" category serving as the reference group. Results were expressed as adjusted odds ratios (AORs) with 95% confidence intervals (CIs), and a p-value of less than 0.05 was considered statistically significant. Geographic distribution of participants by governorate was visualized using OpenStreetMap data [23].

## Ethical considerations

The DHS program provided ethical approval for the JPFHS before the Jordanian Department of Statistics started data collection. Informed consent was a prerequisite for an individual to participate in the survey. The current study constituted a secondary analysis of de-identified data of JPFHS 2023. The DHS program provided the data and approved the research proposal of the present work. Given the anonymized nature of data, no additional ethical approval was required to analyze it.

## Results

Data were collected from 12,304 Jordanian women of reproductive age (20–49 years). Among these, 9,851 women reported that they do not practice BSE, while 2,453 women indicated that they had performed BSE.

The 40–49 age group represented the largest proportion among women who reported practicing BSE (56.5%). The same age group also comprised the highest proportion of non-performers but at a lower rate (38%). Most women in both

groups were married (92.1% and 92.9%) with predominant multiparity. When considering current pregnancy status, the majority of women in both groups reported being either not pregnant or unsure of their pregnancy status.

The proportion of women who completed their higher education was greater among those who perform BSE compared to non-performers (40.5% vs 33.4%); however, in terms of employment, most women in both groups were not currently working.

The distribution of the wealth index revealed notable differences between women who did not perform BSE and those who did. Women who performed BSE were more concentrated in the higher wealth categories, with 24.0% in the richer group and 26.4% in the richest group. In contrast, among women who did not perform BSE, the wealth index was relatively evenly distributed across the poorest (21.2%), poorer (22.1%), and middle (21.3%) categories, with a slight decrease in the richer (18.7%) and richest (16.7%) categories.

BSE rates across different governorates in Jordan reveals notable regional variations (Fig 1). The highest percentage of women reporting BSE performance was observed in Amman Governorate, with 49.1% of women practicing BSE, followed by Zarqa Governorate at 15.2%. In contrast, several governorates showed relatively low BSE rates, such as Mafraq (1.9%), Ma'an (1.1%), and Tafilah (0.8%). Urban residence was predominant in both groups, with 90.7% of women who did not perform BSE and 93% of those who performed BSE living in urban areas. Detailed baseline characteristics are provided in **Table 1**.

In the multivariate analysis (**Table 2**), a clear trend emerged showing increased odds of BSE performance with advancing age. Compared to the reference group (20–24 years), the adjusted odds ratios (AOR) for BSE performance were

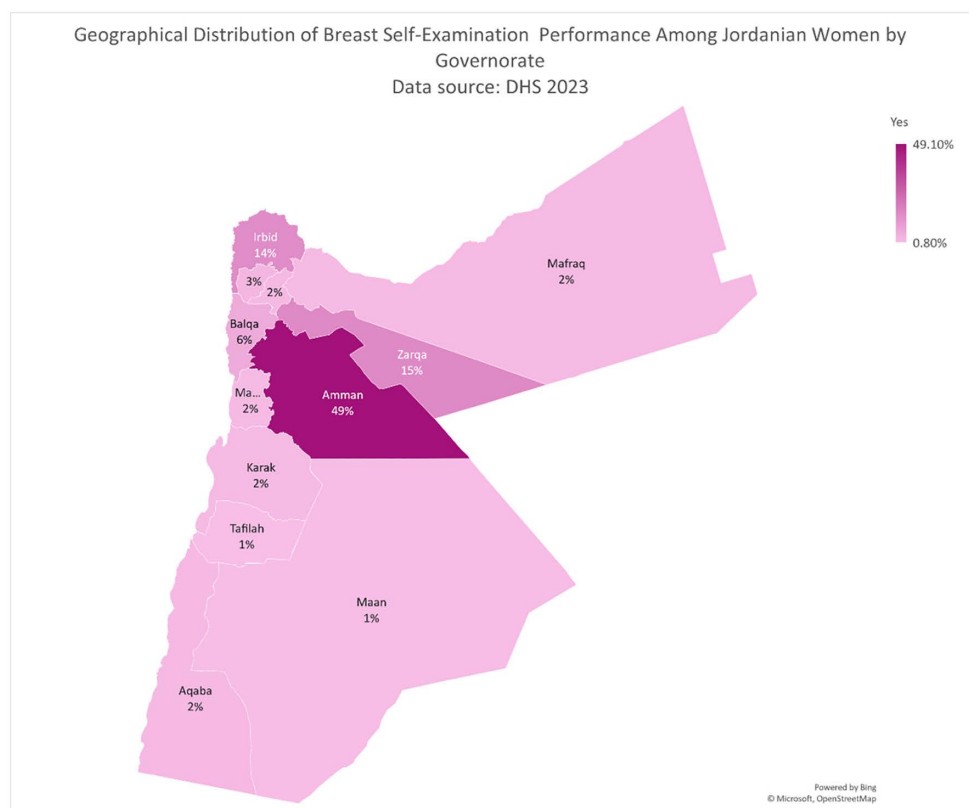

**Fig 1. Geographical Distribution of Breast Self-Examination Performance Among Jordanian Women by Governorate Created Using OpenStreetMap (© OpenStreetMap contributors).**

**Table 2. Multivariate logistic regression results for predictors of BSE.**

| Variables | Parameter estimate | | | | | | |
|---|---|---|---|---|---|---|---|
| | | **B** | **Standard Error** | **AOR** | **95% Confidence Interval for AOR** | | |
| | | | | | **Lower** | **Upper** | **P Value** |
| **Age in 5-year groups** | **20-24** | .000a | | 1 | | | |
| | **25-29** | 0.045 | 0.267 | 1.046 | 0.62 | 1.766 | 0.866 |
| | **30-34** | 0.598 | 0.234 | 1.818 | 1.147 | 2.879 | **0.011** |
| | **35-39** | 0.878 | 0.241 | 2.405 | 1.498 | 3.863 | **< 0.001** |
| | **40-44** | 0.957 | 0.232 | 2.604 | 1.652 | 4.105 | **< 0.001** |
| | **45-49** | 1.123 | 0.237 | 3.076 | 1.93 | 4.9 | **< 0.001** |
| **Body Mass Index** | **Underweight** | .000a | | 1 | | | |
| | **normal weight** | 0.745 | 0.566 | 2.106 | 0.694 | 6.393 | 0.188 |
| | **Overweight** | 0.578 | 0.555 | 1.783 | 0.6 | 5.298 | 0.298 |
| | **Obese** | 0.693 | 0.555 | 2 | 0.673 | 5.949 | 0.212 |
| **Current marital status** | **Married** | .000a | | 1 | | | |
| | **Widowed** | 0.14 | 0.257 | 1.15 | 0.694 | 1.906 | 0.587 |
| | **Divorced** | −0.175 | 0.263 | 0.84 | 0.501 | 1.407 | 0.507 |
| | **No longer living together/separated** | −1.341 | 0.931 | 0.262 | 0.042 | 1.626 | 0.15 |
| **Parity** | **Nulliparous** | .000a | | 1 | | | |
| | **Primiparous** | 0.146 | 0.262 | 1.157 | 0.692 | 1.936 | 0.577 |
| | **Multiparous** | 0.555 | 0.221 | 1.741 | 1.128 | 2.689 | **0.012** |
| **Currently pregnant** | **No or unsure** | .000a | | 1 | | | |
| | **Yes** | −0.048 | 0.228 | 0.954 | 0.609 | 1.492 | 0.835 |
| **Frequency smokes cigarettes** | **Does not smoke** | .000a | | 1 | | | |
| | **Every day** | 0.535 | 0.199 | 1.707 | 1.154 | 2.524 | **0.007** |
| | **Some days** | 0.239 | 0.36 | 1.27 | 0.627 | 2.572 | 0.507 |
| **Highest educational level** | **No education** | .000a | | 1 | | | |
| | **Primary** | 0.556 | 0.425 | 1.745 | 0.757 | 4.019 | 0.191 |
| | **Secondary** | 0.881 | 0.398 | 2.413 | 1.105 | 5.268 | **0.027** |
| | **Higher** | 0.847 | 0.43 | 2.333 | 1.003 | 5.424 | **0.049** |
| **Respondent currently working** | **No** | .000a | | 1 | | | |
| | **Yes** | 0.078 | 0.157 | 1.081 | 0.794 | 1.473 | 0.619 |
| **Wealth index combined** | **Poorest** | .000a | | 1 | | | |
| | **Poorer** | 0.054 | 0.171 | 1.055 | 0.755 | 1.474 | 0.754 |
| | **Middle** | 0.449 | 0.138 | 1.566 | 1.195 | 2.052 | **0.001** |
| | **Richer** | 0.414 | 0.161 | 1.512 | 1.103 | 2.074 | **0.01** |
| | **Richest** | 0.433 | 0.19 | 1.541 | 1.062 | 2.237 | **0.023** |
| **Frequency of using internet last month** | **Not at all** | .000a | | 1 | | | |
| | **Less than once a week** | 0.684 | 0.405 | 1.982 | 0.896 | 4.383 | 0.091 |
| | **At least once a week** | 0.451 | 0.284 | 1.569 | 0.898 | 2.743 | 0.113 |
| | **Almost every day** | 0.371 | 0.154 | 1.45 | 1.072 | 1.961 | **0.016** |
| **Frequency of watching television** | **Not at all** | .000a | | 1 | | | |
| | **Less than once a week** | 0.078 | 0.163 | 1.081 | 0.785 | 1.49 | 0.631 |
| | **At least once a week** | −0.152 | 0.128 | 0.859 | 0.668 | 1.104 | 0.235 |

*(Continued)*

**Table 2.** (Continued)

| Variables | Parameter estimate | | | | | | |
|---|---|---|---|---|---|---|---|
| | | B | Standard Error | AOR | 95% Confidence Interval for AOR | | P Value |
| | | | | | Lower | Upper | |
| Frequency of listening to radio | Not at all | .000a | | 1 | | | |
| | Less than once a week | 0.064 | 0.151 | 1.066 | 0.792 | 1.434 | 0.675 |
| | At least once a week | 0.002 | 0.157 | 1.002 | 0.736 | 1.365 | 0.989 |
| Frequency of reading newspapers or magazines | Not at all | .000a | | 1 | | | |
| | Less than once a week | 0.56 | 0.183 | 1.751 | 1.223 | 2.506 | **0.002** |
| | At least once a week | 0.407 | 0.183 | 1.503 | 1.049 | 2.152 | **0.026** |
| Type of place of residence | Urban | .000a | | 1 | | | |
| | Rural | −0.012 | 0.159 | 0.988 | 0.723 | 1.349 | 0.938 |
| Governates | Amman | .000a | | 1 | | | |
| | Balqa | 0.371 | 0.165 | 1.449 | 1.048 | 2.003 | **0.025** |
| | Zarqa | 0.316 | 0.181 | 1.372 | 0.962 | 1.955 | 0.081 |
| | Madaba | 0.101 | 0.196 | 1.107 | 0.754 | 1.624 | 0.604 |
| | Irbid | −0.436 | 0.18 | 0.647 | 0.454 | 0.92 | **0.016** |
| | Mafraq | −0.523 | 0.245 | 0.593 | 0.367 | 0.958 | **0.033** |
| | Jarash | −0.002 | 0.206 | 0.998 | 0.666 | 1.495 | 0.993 |
| | Ajloun | 0.987 | 0.2 | 2.682 | 1.811 | 3.971 | >0.001 |
| | Karak | −0.096 | 0.24 | 0.909 | 0.567 | 1.457 | 0.691 |
| | Tafiela | −0.137 | 0.223 | 0.872 | 0.563 | 1.349 | 0.537 |
| | Ma'an | −0.052 | 0.298 | 0.949 | 0.529 | 1.704 | 0.861 |
| | Aqaba | 0.647 | 0.226 | 1.911 | 1.226 | 2.978 | **0.004** |

Abbreviations: AOR = Adjusted odds ratio.

significantly higher in older age groups: 30–34 years (AOR: 1.818; 95% CI: 1.147–2.879; p = 0.011), 35–39 years (AOR: 2.405; 95% CI: 1.498–3.863; p < 0.001), 40–44 years (AOR: 2.604; 95% CI: 1.652–4.105; p < 0.001), and 45–49 years (AOR: 3.076; 95% CI: 1.930–4.900; p < 0.001). Notably, the age group 25–29 years (AOR: 1.046; 95% CI: 0.620–1.766; p = 0.888) did not exhibit a significant association with BSE performance.

Educational level demonstrated a significant association with BSE performance, particularly among women with secondary and higher education. Compared to women with no formal education, those with secondary education were more than twice as likely to perform BSE (AOR: 2.413; 95% CI: 1.105–5.268; p = 0.027). Similarly, women with higher education showed significantly increased odds of BSE performance (AOR: 2.333; 95% CI: 1.000–5.424; p = 0.049). In contrast, women with primary education did not exhibit a significant association (AOR: 1.745; 95% CI: 0.757–4.019; p = 0.191).

Wealth index showed a clear trend of increasing odds of BSE performance as wealth status improved. Compared to the poorest group, women in the middle (AOR: 1.566; 95% CI: 1.195–2.052; p = 0.001), richer (AOR: 1.512; 95% CI: 1.103–2.074; p = 0.010), and richest quintiles (AOR: 1.541; 95% CI: 1.062–2.237; p = 0.023) demonstrated significantly higher odds of BSE performance.

Parity also played a role, with multiparous women showing significantly higher odds of performing BSE compared to nulliparous women (AOR: 1.741; 95% CI: 1.128–2.689; p = 0.012). Smoking behavior was another significant factor, as women who smoked every day were more likely to perform BSE than non-smokers (AOR: 1.707; 95% CI: 1.154–2.524; p = 0.007).

In terms of media consumption, frequent internet use was associated with higher BSE performance. Women using the internet almost daily had significantly greater odds of performing BSE compared to those not using the internet at all (AOR: 1.450; 95% CI: 1.072–1.961; p = 0.016). Similarly, reading newspapers or magazines was positively associated with BSE, with women reading less than once a week (AOR: 1.751; 95% CI: 1.223–2.506; p = 0.002) or at least once a week (AOR: 1.503; 95% CI: 1.049–2.152; p = 0.026) showing increased odds of performance.

Regional differences were also evident. Women in Ajloun (AOR: 2.682; 95% CI: 1.811–3.971; p < 0.001), Aqaba (AOR: 1.911; 95% CI: 1.226–2.978; p = 0.004), and Balqa (AOR: 1.449; 95% CI: 1.048–2.003; p = 0.025) were significantly more likely to perform BSE compared to those in Amman. Conversely, women in Irbid (AOR: 0.647; 95% CI: 0.454–0.920; p = 0.016) and Mafraq (AOR: 0.593; 95% CI: 0.367–0.958; p = 0.033) showed significantly lower odds of BSE performance. No significant associations were observed for other governorates.

Our analysis revealed no significant associations between performing BSE and other factors, including current marital status, type of place of residence, BMI, current pregnancy status, employment status, frequency of watching television, or frequency of listening to the radio.

## Discussion

This study represents the first effort to examine the factors influencing BSE performance among adult women in Jordan using nationally representative data of 2023. Jordan is a country where breast cancer constitutes the most common cancer among females and the leading cause of cancer-related mortality in females [5]. Our analysis demonstrated a strong association between age and BSE performance, with older women significantly more likely to perform BSE. Women aged 45–49 were found to be three times more likely to practice BSE compared to those aged 20–24. These findings are consistent with studies by Al-Rifai et al. in Jordan 2015 [18], Okyere et al. in Namibia 2023 [20], and Wassie et al. in Kenya 2024 [24]. This trend could be attributed to older women being more prone to health issues and more frequent interactions with healthcare providers [25].

The current analysis included data retrieved from the most updated JPFHS which allowed to investigate the long-term effect of the JBCP on the practice of BSE among Jordanian women [9]. Among the studied independent variables, media-related variables were considered, including internet and television, which provides novel insights on the effect of digital and televised campaigns which was previously underexplored.

The analysis highlighted a positive correlation between educational attainment and BSE practice, with higher education levels being associated with increased BSE performance. These findings are consistent with previous studies, including Al-Rifai et al. in Jordan 2015 [18], Al-Rifai et al. in Egypt 2017 [26], Okyere et al. in Namibia 2023 [20], and Wassie et al. in Kenya 2024 [24]. Educated women are more likely to have access to health information [27], enhancing their awareness of breast cancer and the importance of early detection methods like BSE. This increased awareness can lead to a greater understanding of the benefits of regular BSE, thereby promoting its practice. Furthermore, educated women may have better access to healthcare services and resources [27], providing more opportunities to learn about and practice BSE.

The analysis of the wealth index and its association with BSE performance revealed a significant increase in performance as wealth levels increased. This finding aligns with previous studies, including those by Al-Rifai et al. in Jordan 2015 [18], Okyere et al. in Namibia 2023 [20], and Wassie et al. in Kenya 2024 [24]. This trend may be attributed to increased access to healthcare resources, educational opportunities, and health information among wealthier women, which collectively enhance their awareness and practice of preventive health measures like BSE.

The study revealed a pro-rich bias in breast screening rates, with women in lower socioeconomic strata less likely to access early cancer detection programs. Despite improvements in Jordan's socioeconomic indicators—such as a rise in per capita GDP from $4,289 in 2005 to $10,071 in 2019 [28], and a decline in maternal mortality from 96 to 41 per 100,000 live births [29]—inequalities in breast screening persist.

To address lower BSE rates among disadvantaged women, targeted interventions are essential. Educational campaigns tailored to these groups should emphasize the importance of BSE and early detection, with community health workers and local organizations aiding outreach efforts. Increasing access to screening services, such as mobile mammography units provided by the JBCP, can effectively reach underserved areas [30]. Financial barriers must also be tackled through collaborations offering free or subsidized services, like Anera's program, which connects women to free mammograms at Al-Hilal Hospital in Eastern Amman [31].

Regarding parity, our study found that multiparous women were associated with increase in BSE performance which align with Al-Rifai et al. in Jordan 2015 [18], Okyere et al. in Namibia 2023 [20], and Wassie et al. in Kenya 2024 [24]. The frequent interactions with healthcare services during multiple pregnancies may contribute to increased health awareness. Regular contact with healthcare providers offers opportunities for education on breast health, emphasizing the importance of early detection methods like BSE. This correlation highlights the importance of utilizing maternal healthcare visits to educate women about breast health [32]. By integrating breast cancer awareness and BSE training into prenatal and postnatal care, healthcare providers can enhance early detection efforts.

Our analysis of smoking behavior revealed a significant association between higher smoking frequency and increased performance of BSE. This finding is consistent with previous research exploring the relationship between smoking and health practices [33]. For example, a study on acculturation among African American women identified a correlation between smoking and adherence to BSE frequency guidelines, suggesting that lifestyle factors may influence engagement in preventive health behaviors [33]. One possible explanation for this association is that smokers might exhibit heightened health awareness, leading them to monitor their health more closely due to the known risks of smoking. However, future researchers are encouraged to further research the effect of smoking status on the attitudes toward BSE to provide more robust evidence.

Our analysis identified a positive correlation between media consumption and BSE performance, specifically with increased frequency of internet use and reading newspapers or magazines. Women who engaged more frequently with these media were more likely to perform BSE, which align with Al-Rifai et al. in Jordan 2015 [26] and contrast Okyere et al. in Namibia 2023 [20]. In contrast, watching television and listening to the radio showed no significant association with BSE performance which align with Al-Rifai et al. in Jordan 2015 [18] and Okyere et al. in Namibia 2023 [20]. These findings suggest that the type of media consumed plays a crucial role in influencing health-related behaviors. Public media can effectively disseminate health warnings and provide information on available clinical and laboratory testing services. However, with less than 30% of women accessing newspapers or magazines, relying solely on these platforms may not substantially increase screening rates [34].

In contrast, over 70% of women reported daily internet use, presenting a valuable opportunity to raise awareness, particularly among younger women [34]. Digital platforms can be utilized through social media campaigns, educational videos, and interactive content to promote breast cancer screening and self-examination [35]. Integrating internet-based strategies with traditional media can expand outreach and enhance awareness across different age groups, ensuring a more comprehensive approach to health education.

We found no significant association between the type of residence (urban or rural) and BSE practices, which contrasts with the findings of Al-Rifai et al. in Jordan 2015 [18], who reported a decrease in BSE practices in rural areas. This suggests that efforts to improve health education, raise awareness, and enhance access to healthcare services in rural areas may have been effective in narrowing the gap in BSE practices between urban and rural populations.

Our analysis identified regional variations in BSE practices among women. Women residing in Balqa, Ajloun, and Aqaba demonstrated significantly higher BSE performance compared to those in Amman. Conversely, women from Irbid and Mafraq exhibited significantly lower BSE performance. No significant differences in BSE practices were observed among women in other governorates. These variations may be influenced by factors such as differences in health education outreach, cultural norms, socioeconomic status, and accessibility to healthcare services across regions. The

nationwide efforts of the JBCP were intended to eliminate the geographical disparity, which has persisted since 2007 [36]. Such disparities can manifest as lower healthcare infrastructure and fewer educational initiatives. Addressing these disparities requires tailored public health strategies that consider the unique characteristics and needs of each region to enhance BSE awareness and practice uniformly across the country. The efforts of the JBCP in eliminating healthcare-related disparities are manifested as a lack of significant associations between urban or rural residence and BSE practice. The findings of the current study can be considered as a basis to focus on governates that showed lower BSE practice. We also recommend conducting further research to gain a deeper understanding of the variations in BSE practice across different regions.

Several factors, such as BMI, residence type, marital status, employment status, and pregnancy history were not significantly associated with BSE in the final regression model. BMI showed non-significant association with BSE practice which suggests that body weight does not influence women attitude toward BSE. The non-significant associations with marital status, pregnancy, and employment may indicate that BSE practice is driven by individual awareness and education rather than these variables. Moreover, the non-significant association between watching television or listening to the radio and performing BSE can be explained by the increasing reliance on digital media.

## Strengths and limitations

A major strength of our study is the large sample size, which included 12,304 Jordanian women of reproductive age. The use of data from the reliable and well-established Demographic and Health Survey adds credibility to our findings. By employing multivariate analysis, we were able to identify a variety of factors that influence BSE practices, providing a comprehensive understanding of the factors contributing to BSE behavior. The study also highlights regional variations in BSE practices, offering crucial insights into the need for region-specific interventions. Furthermore, the inclusion of wealth status and inequality helped us to identify socioeconomic and geographical factors associated with low cancer screening rates. To minimize recall bias, the study focused on BSE practices within the past 12 months, ensuring more accurate reporting.

However, there are several limitations to consider. As a cross-sectional study, it can only identify associations, not causal relationships. Self-reported BSE practices may be affected by recall or social desirability bias. Individual-level factors, such as access to healthcare, prior exposure to health education campaigns, family history of breast cancer, awareness of warning signs, and cultural beliefs, were not accounted for, which may limit insights into regional disparities. Cultural barriers and reluctance to discuss sensitive health issues could have introduced response bias, potentially varying between urban and rural areas. Women exposed to health education campaigns may have been more likely to respond, suggesting potential self-selection bias. Additionally, multicollinearity among independent variables, such as education level and wealth index, was not formally assessed, which could have influenced regression estimates. Future studies should address these issues to strengthen the robustness of findings.

## Recommendations

Based on this study, several recommendations can enhance BSE among Jordanian women. First, younger women, particularly those aged 25–29, should be targeted in health interventions to encourage early BSE adoption. Educational campaigns should focus on women with lower education levels, as those with secondary or higher education were more likely to perform BSE. Improving literacy and education, especially for those with no formal or primary education, could help increase BSE practice. Additionally, socioeconomic factors like wealth impact BSE rates, so awareness programs should target economically disadvantaged women.

Outreach to nulliparous women could help bridge the gap in BSE performance, as multiparous women showed higher adoption rates. We encourage healthcare providers to educate their patients on BSE practice and provide guidance for them during their interactions with the healthcare system, particularly routine check-ups and family planning visits.

Utilizing media, especially the internet and newspapers, can increase BSE awareness, particularly for digitally connected women. However, conventional community-based health education campaigns remain necessary to outreach older women and those who reside rural areas since these two populations may have limited access to digital platforms.

Lastly, tailored interventions in regions like Irbid and Mafraq, where BSE rates are lower, could address cultural and regional differences through localized awareness campaigns. The JBCP initiatives can play a major role in tailoring such interventions; however, community-based organizations and non-governmental organizations (NGOs) can also have significant contributions to this field. Primary healthcare centers in these areas should be advised to strengthen breast cancer health education initiatives to increase awareness about screening, including BSE.

## Conclusion

Older women, those with higher levels of education, and women with greater wealth were more likely to perform BSE. Multiparous women, daily smokers, and those who frequently use the internet or regularly read newspapers and magazines also reported higher rates of BSE. Regional differences emerged: women in Ajloun, Aqaba, and Balqa were more likely to perform BSE, while rates were lower in Irbid and Mafraq. These findings highlight the need for targeted interventions aimed at younger women, those with lower education, and individuals in economically disadvantaged areas to improve BSE uptake and promote early breast cancer detection. Further research is needed to explore the cultural and behavioral barriers that may influence BSE practices.

## Author contributions

**Conceptualization:** Amr Ahmed Aly Ibrahim.

**Formal analysis:** Amr Ahmed Aly Ibrahim.

**Investigation:** Amr Ahmed Aly Ibrahim.

**Methodology:** Amr Ahmed Aly Ibrahim.

**Supervision:** Mahmoud Shaaban Abdelgalil.

**Writing – original draft:** Sara Hosny El-Farargy, Nour Eldein Saad, Aya Khafage.

**Writing – review & editing:** Moaz Yasser Darwish, Mahmoud Shaaban Abdelgalil.

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
