## [Decision Letter · Decision Letter 0]

21 Jun 2025

PONE-D-25-00461Exploring Socio-Demographic Determinants of Breast Self-Examination Practices Among Jordanian Women: Insights from the 2023 Population-Based SurveyPLOS ONE

Dear Dr. Shaaban Abdelgalil,

Thank you for submitting your manuscript to PLOS ONE. After careful consideration, we feel that it has merit but does not fully meet PLOS ONE’s publication criteria as it currently stands. Therefore, we invite you to submit a revised version of the manuscript that addresses the points raised during the review process.

We look forward to receiving your revised manuscript.

Kind regards,

Nülüfer Erbil, Ph.D, Prof.

Academic Editor

PLOS ONE

Journal Requirements:

2. Please amend your list of authors on the manuscript to ensure that each author is linked to an affiliation. Authors’ affiliations should reflect the institution where the work was done (if authors moved subsequently, you can also list the new affiliation stating “current affiliation:….” as necessary).

Reviewers' comments:

Reviewer's Responses to Questions

**Comments to the Author**

1. Is the manuscript technically sound, and do the data support the conclusions?

Reviewer #1: Yes

Reviewer #2: Partly

Reviewer #3: Yes

2. Has the statistical analysis been performed appropriately and rigorously? 

Reviewer #1: Yes

Reviewer #2: Yes

Reviewer #3: Yes

3. Have the authors made all data underlying the findings in their manuscript fully available?

Reviewer #1: Yes

Reviewer #2: Yes

Reviewer #3: Yes

4. Is the manuscript presented in an intelligible fashion and written in standard English?

Reviewer #1: Yes

Reviewer #2: Yes

Reviewer #3: Yes

5. Review Comments to the Author

Reviewer #1: Exploring Socio-Demographic Determinants of Breast Self-Examination Practices Among

Jordanian Women: Insights from the 2023 Population-Based Survey

Strengths of the Manuscript

Strengths of the Manuscript

1. Timely and Relevant Topic

o Breast cancer is a significant public health concern, and the study provides valuable information regarding determinants of BSE practice in Jordan.

o The focus on socio-demographic determinants is relevant to informing intervention design and increasing screening behavior.

Large Sample Size

o The study has the strength of having a nationally representative sample (12,304 married women) that ensures generalizability of findings

2. Use of JDHS Data

o The study is founded on a well-established and trustworthy dataset, contributing to the validity of its findings.

3. Multivariate Analysis

o The use of logistic regression is appropriate for the examination of different factors influencing BSE practices.

Areas for Improvement

1. Methodological Concerns

• Cross-Sectional Design Limitation

o The study relies on a cross-sectional dataset, and causal inferences are partial. This must be articulated more explicitly under the limitations section.

• Potential Biases in Self-Reported Data

o Since BSE practices were self-reported by the participants, recall bias and social desirability bias may affect response validity. Discussion of validation measures or consideration of this limitation more explicitly would strengthen the study

• Confounding Variables

o While adjustments were made for several socio-demographic and behavioral variables, other possible confounders such as family history of breast cancer, access to health care, and cultural beliefs were not considered. Adjustment for these variables may strengthen the findings.

2. Clarity in Data Presentation

• Table Organization and Interpretation

o The paper presents a great deal of statistical data, yet some tables and results must be more clearly organized for readability.

o Consider bolding significant findings or summarizing the most critical points in a discussion section

• Multicollinearity Considerations

o Some of the independent variables, such as education level and wealth index, are very likely to be highly correlated. It would strengthen the analysis to indicate whether multicollinearity diagnostics were conducted.________________________________________

3. Discussion and Interpretation of Findings

• Comparisons with Previous Research

o While the paper refers to earlier studies, there is room to compare more intently with similar research done elsewhere beyond Jordan to place findings

• Cultural and Behavioral Explanations

o The paper mentions cultural barriers to BSE uptake but does not discuss at length specific cultural beliefs that may discourage self-examination. Further explanation on this issue would provide more insight

• Implications for Public Health Interventions

o Although recommendations are given, real-world applications to healthcare providers, policymakers, and NGOs in Jordan must be more clearly outlined

4. Ethical Considerations

• Ethical Approval and Consent

o The study used secondary JDHS data that is in the public domain. However, a more explicit explanation of ethical clearance procedures followed by the JDHS would facilitate transparency

Recommendations for Improvement

1. Strengthen the discussion on limitations, particularly for cross-sectional design and self-reporting data.

2. Describe methodological details, e.g., how multicollinearity was accounted for in regression models.

3. Discuss regional and cultural determinants of BSE practices to provide more meaningful contextual information.

4. Provide easier-to-read tables and figures to make data interpretation easier.

5. Make recommendations stronger by providing concrete actionable steps for healthcare providers and policymakers.

Conclusion

The manuscript makes an important contribution to the understanding of BSE practice among Jordanian women. Nevertheless, the resolution of methodological concerns, clarification of data presentation, and enhancement of cultural and policy-related discussions will substantially strengthen the manuscript's clarity and contribution.

Final Recommendation: Major revisions required before acceptance for publication in PLOS ONE.

Reviewer #2: Dear authors

I congratulate you on your work. Although it is written in a clear and understandable language, the subject discussed in the article is not a current issue. There are many studies in the literature on the subject. It would be more appropriate to publish it in a local journal.

Best regards

Reviewer #3: This manuscript addresses an important public health issue by examining the socio-demographic factors associated with breast self-examination (BSE) practices among Jordanian women using nationally representative data from the 2023 Jordan Population and Family Health Survey. The topic is timely and relevant, particularly in the context of early breast cancer detection strategies in middle-income countries.

The study benefits from a large sample size, appropriate statistical methods, and a clear presentation of findings. The manuscript is generally well organized, with adequate linkage to existing literature and a focus on practical recommendations. However, certain sections would benefit from improved clarity, more detailed interpretation of unexpected or non-significant findings, and slight enhancements in language and structure. Additionally, expanding the discussion around the implications of the results—particularly in relation to healthcare access, media outreach, and regional disparities—would strengthen the overall impact of the paper.

With minor to moderate revisions, this study has the potential to contribute meaningful insights to the field of cancer prevention and women’s health in the Middle East.

Abstract

The abstract effectively presents the study’s background, methods, key results, and conclusions. It communicates the public health relevance of BSE practices and offers clear insights into socio-demographic disparities. To further enhance its impact, consider including one or two statistical values to support your findings, revising minor grammatical issues, and clarifying the specific statistical method used.

While the methods section of the abstract provides sufficient information about the dataset and sample characteristics, please consider briefly mentioning the specific statistical method used (e.g., logistic regression), as this strengthens methodological transparency.

The results are clearly presented and aligned with the study’s objectives. Nevertheless, including at least one adjusted odds ratio (AOR) and p-value in the abstract would enhance the precision and credibility of the reported findings.

The conclusion effectively highlights the importance of targeted interventions. You may consider briefly indicating potential implications for policy or future research directions in a concise sentence.

The language is generally clear and concise. However, a few phrases could be revised for improved grammar and readability (e.g., ‘frequent reading newspaper of magazines’ → ‘frequent reading of newspapers or magazines’). A light copy-editing pass is recommended.

Introduction

The introduction provides a solid overview of the global and national burden of breast cancer and outlines the importance of BSE among Jordanian women. To further strengthen this section, I recommend clarifying the rationale for focusing on BSE despite evolving international recommendations, elaborating on how the present study differs from previous DHS-based analyses, and thematically organizing the discussion of barriers. These changes would improve the logical flow and enhance the justification for the study’s aims.

Lines 1–6

The manuscript opens with strong global breast cancer statistics, which provide important context. However, I recommend transitioning more swiftly from the global burden to the national (Jordanian) context to better anchor the reader in the geographic scope of the study.

Lines 7–12

The section effectively presents the prevalence and mortality rates of breast cancer in Jordan. To enhance its impact, consider briefly indicating whether national screening efforts (e.g., JBCP) have shown any progress in early diagnosis or survival rates since their implementation.

Lines 21–30

While it is valid to highlight the affordability and accessibility of BSE, it should be noted that BSE is not strongly recommended in recent international screening guidelines. The authors are encouraged to justify their focus on BSE either by referencing Jordan-specific guidelines or explaining population-level barriers that make BSE particularly relevant in this setting.

Lines 12–22, Page 4

The manuscript refers to Al-Rifai et al. (2012) to establish a gap in the literature. While this is appreciated, it would strengthen the argument to more clearly articulate what this study offers that previous work has not—such as the inclusion of behavioral or media consumption variables, or an emphasis on regional disparities.

Lines 3–11, Page 4

The discussion of barriers—including cultural, religious, and economic factors—is comprehensive. To improve readability, consider organizing these barriers thematically (e.g., structural, informational, cultural) rather than listing them consecutively.

Materials and Methods Section

The Materials and Methods section is generally comprehensive and appropriate for the study design. To enhance transparency and reproducibility, I recommend the following: (1) add details on the sampling design and timeframe of data collection; (2) clarify the exclusion of unmarried women; (3) improve justification for variable categorization; (4) elaborate on weighting and model diagnostics; and (5) include a brief ethical statement confirming secondary use of Demographic and Health Surveys (DHS) data. These additions will strengthen the methodological rigor and reader comprehension of the study.

Lines 4–6

The authors state that the study utilized data from the 2023 Jordan Population and Family Health Survey (JPFHS). While this is appropriate, I recommend including the date of data collection (e.g., months/season) and a brief description of the sampling design (e.g., stratified, two-stage cluster sampling) for transparency and replicability.

Lines 7–11

The definition of the dependent variable is clear; however, the phrase “breast cancer screening” may be misleading since the outcome is specifically related to Breast Self-Examination (BSE), not clinical or imaging-based screening. Please revise this terminology for precision.

Lines 12–19

The inclusion and exclusion criteria are appropriate. That said, the rationale for limiting the sample to ever-married women could be briefly explained. For example, were unmarried women excluded due to data availability, cultural reasons, or screening relevance? This clarification would help readers understand potential biases in sample selection.

Lines 20–30

The range and categorization of independent variables are comprehensive and justified. However, some categorization decisions (e.g., for internet use and BMI) could benefit from referencing previous studies or DHS methodology to support the choices made.

Lines 14–25

The use of weighted logistic regression is appropriate for DHS data. The description of weighting is technically accurate but could be made clearer for broader audiences. I suggest briefly explaining why weighting is necessary in DHS analysis and specifying which weight variable was used.

Also, it would be helpful to mention whether any multicollinearity diagnostics were performed before multivariable modeling, and whether model fit or goodness-of-fit statistics were assessed.

Ethical Considerations

Although the authors mention that the data is publicly available, it would be beneficial to explicitly state that ethical approval and informed consent were handled by the original DHS program, and that no additional ethical clearance was required for secondary analysis.

Results

The descriptive findings are informative; however, the presentation would benefit from a more structured flow. The transition between different variable groups (e.g., socio-demographic, behavioral, regional) appears abrupt. Grouping these variables more clearly in the narrative would enhance readability and comprehension for the reader.

Table 1:

While Table 1 provides a comprehensive overview of the sample characteristics, the accompanying text primarily reiterates values already shown in the table. The narrative should instead emphasize notable trends or meaningful contrasts that may not be immediately evident from the table alone. This would reduce redundancy and increase the analytical value of the results section.

Multivariate Analysis – Reporting of Significant and Non-significant Results:

The logistic regression results are well reported, with appropriate use of adjusted odds ratios and confidence intervals. However, the text focuses predominantly on statistically significant findings, while non-significant associations are either overlooked or insufficiently addressed. Reporting both significant and non-significant outcomes strengthens transparency and helps to avoid potential bias in interpretation.

Visual Data (Figure 1):

The inclusion of a geographical distribution figure adds value to the manuscript. However, the figure is underutilized in the text. A more detailed interpretation of the observed regional differences and their potential public health implications would enhance the overall narrative and contextual relevance.

Discussion

The authors present a good summary of the main results at the beginning of the discussion. However, it would be helpful to briefly restate why these findings are important for public health in Jordan.

The manuscript compares its findings with previous studies from Jordan and other countries, which is appreciated. The discussion could be improved by explaining more clearly how this study adds new knowledge to the existing literature.

Some results, such as the positive relationship between smoking and BSE, are surprising. The discussion mentions a possible reason, but it would be helpful to say that more research is needed to better understand this finding.

The authors found that BSE practices vary across regions in Jordan. This is important. However, the reasons for these differences are not fully explained. A short paragraph discussing possible causes (e.g., differences in health education or access to care) would be useful.

Some factors (e.g., BMI, place of residence, television watching) were not related to BSE, but the discussion does not mention them. Even if they are not significant, it is good to briefly comment on why this might be the case.

Strength and Limitation

The strengths of the study are clearly described, particularly the large sample size and the use of a reliable national dataset. These aspects support the generalizability of the findings.

The use of multivariate analysis and inclusion of regional and socioeconomic factors are commendable and contribute to the study’s comprehensiveness.

The limitations section identifies relevant issues such as the cross-sectional design and self-reported nature of BSE data. However, it would be useful to expand on the limitation that the study did not include some potentially important individual-level variables, such as:

family history of breast cancer,

knowledge of breast cancer warning signs,

access to screening facilities,

prior participation in health education programs.

The mention of cultural barriers is appropriate, but a clearer explanation of how such factors may vary by region (or urban/rural residence) would make this limitation more meaningful.

Recommendation

The recommendations are generally relevant and directly linked to the findings. The focus on younger, less educated, and economically disadvantaged women is appropriate.

The recommendation to use digital media and newspapers is logical; however, it might be useful to add a brief note on the accessibility of these media among older or rural women.

Outreach to nulliparous women is a valuable point, and it may be strengthened by briefly mentioning how health providers (e.g., during family planning or general check-ups) could play a role in this outreach.

The call for region-specific awareness efforts is important. This could be improved by briefly suggesting which stakeholders (e.g., local health departments, NGOs) could help implement such programs.

Conclusion

The conclusion accurately reflects the key findings of the study and provides a concise summary.

However, it would be helpful to rephrase the sentence: “frequent reading newspaper of magazines” to “frequent reading of newspapers or magazines” for clarity and correctness.

While the statistical findings are summarized, the conclusion may be slightly strengthened by including one sentence on the potential public health value of improving BSE practices (e.g., increased early detection, reduced late-stage diagnosis).

Ending with a brief call for future research—such as follow-up studies, interventions, or qualitative exploration of cultural barriers—would add depth and signal future directions.

6. PLOS authors have the option to publish the peer review history of their article (what does this mean?). If published, this will include your full peer review and any attached files.

Reviewer #1: No

Reviewer #2: No

Reviewer #3: No

---

## [Author Response · Author response to Decision Letter 1]

26 Nov 2025

Dear Editor/s

We sincerely thank the editor(s) and reviewers for their valuable and thoughtful insights regarding our research. We believe that the revision process helped us to enhance the clarity of our message and strengthened the overall quality of the manuscript. All comments have been carefully addressed in the following point-by-point responses.

Academic Editor comments

Comment: Please ensure that your manuscript meets PLOS ONE's style requirements, including those for file naming.

Reply: We understand the importance of adhering to the journal’s style requirements. Accordingly, we carefully revised our manuscript in line with PLOS ONE’s guidelines and made the necessary adjustments to ensure full compliance. We also reviewed all file names to confirm that they follow the required formatting

Comment: Please amend your list of authors on the manuscript to ensure that each author is linked to an affiliation. Authors’ affiliations should reflect the institution where the work was done (if authors moved subsequently, you can also list the new affiliation stating “current affiliation:….” as necessary)

Reply: We thank the editor for the clarification. We have carefully amended the authors list to ensure that it follows the mentioned criteria. All authors are affiliated with the institutions where the work was done, and no recent changes in their affiliations have occurred.

Comment: We note that Figure 1 in your submission contain [map/satellite] images which may be copyrighted.

Reply: We acknowledge the editor’s concerns regarding copyright issues for Figure 1. OpenStreetMap permits the public to copy, distribute, transmit, and adapt its data, provided that credit is given to OpenStreetMap and its contributors. In addition to including the required credit on the figure, we have revised the methods section and figure legend to explicitly reference OpenStreetMap as the source.

Reviewer 1

Comment: Cross-Sectional Design Limitation: The study relies on a cross-sectional dataset, and causal inferences are partial. This must be articulated more explicitly under the limitations section.

Reply: We truly appreciate the provided assessment and we would like to thank the reviewer for the effort and time spent to provide such extensive assessment. We acknowledge the reviewer comment and we understand that the current design does not provide causal relationships which is explicitly described in the limitations section:

“Being a cross-sectional study, it can only establish associations rather than causal links.”

Comment: Potential Biases in Self-Reported Data: Since BSE practices were self-reported by the participants, recall bias and social desirability bias may affect response validity. Discussion of validation measures or consideration of this limitation more explicitly would strengthen the study.

Reply: We thank the reviewer for this comment and we agree that the dataset was subjected to recall bias since BSE practices were self-reported. Therefore, recall bias is explicitly mentioned in the limitations section:

“Additionally, self-reported data on BSE practices may be subject to recall and social desirability biases. “

Comment: Confounding Variables: While adjustments were made for several socio-demographic and behavioral variables, other possible confounders such as family history of breast cancer, access to health care, and cultural beliefs were not considered. Adjustment for these variables may strengthen the findings

Reply: We appreciate the reviewer comment regrading the need to consider more confounding variables. However, since the available data was limited, we revised and edited the limitations section to discuss potential factors that could be considered confounders.

“The study also did not account for individual-level factors, such as personal access to healthcare services and screening facilities, the quality of health education campaigns and previous contribution to health education campaigns. which could provide deeper insights into the regional disparities in BSE practices. Other factors that were not considered in the current study include family history of breast cancer, awareness of breast cancer warning signs and cultural beliefs.”

Comment: The paper presents a great deal of statistical data, yet some tables and results must be more clearly organized for readability.

Reply: We thank the reviewer for pointing out the need to reorganize some results and tables considering better readability. We revised and edited our results section to offer a more logical flow. The descriptive findings were revised to start with age and progresses through marital status, parity, and pregnancy status, which are demographic and reproductive characteristics. Education and employment were then summarized followed by wealth to emphasize socioeconomic status. Regional and urban/rural variables were finally described to offer the geographic context. Tables 1 and 2 were also reorganized as, individual-level variables followed reproductive characteristics. Next, educational level, employment and wealth statuses. Then, media exposure took place and finally, regional variables were employed.

Comment: Consider bolding significant findings or summarizing the most critical points in a discussion section.

Reply: We thank the reviewer for this comment, as we believe it is important for scientific research reports to be clear and easy to read. The significant p-values are already highlighted in bold in Table 2. In addition, the discussion section has been revised to ensure that the most important findings are presented at the beginning, followed by a clearer explanation of what this manuscript adds to the existing literature:

“This study represents the first effort to examine the factors influencing BSE performance among adult women in Jordan using nationally representative data of 2023. Jordan is a country where breast cancer constitutes the most common cancer among females and the leading cause of cancer-related mortality in females [5]. Our analysis demonstrated a strong association between age and BSE performance, with older women significantly more likely to perform BSE. Women aged 45–49 were found to be three times more likely to practice BSE compared to those aged 20–24. These findings are consistent with studies by Al-Rifai et al. in Jordan 2015[18], Okyere et al. in Namibia 2023[20], and Wassie et al. in Kenya 2024[26]. This trend could be attributed to older women being more prone to health issues and more frequent interactions with healthcare providers[27].

The current analysis included data retrieved from the most updated JPFHS which allowed to investigate the long-term effect of the JBCP on the practice of BSE among Jordanian women [9]. Among the studied independent variables, media-related variables were considered, including internet and television, which provides novel insights on the effect of digital and televised campaigns which was previously underexplored.”

Comment: Multicollinearity Considerations: Some of the independent variables, such as education level and wealth index, are very likely to be highly correlated. It would strengthen the analysis to indicate whether multicollinearity diagnostics were conducted.

Reply: We thank the reviewer for highlighting this important point. We did not perform formal multicollinearity diagnostics in this analysis. We acknowledge that some variables, such as education level and wealth index, may be correlated, which could influence regression estimates. This has been added to the Discussion as a limitation.

Comment: While the paper refers to earlier studies, there is room to compare more intently with similar research done elsewhere beyond Jordan to place findings

Reply: We thank the reviewer for this comment and we acknowledge the importance of comparing results from Jordan to findings beyond Jordan, especially other countries with low-to-middle income. In fact, one advantage of DHS-based studies is that it follows standard methods that would allow for caparisons with previous or future research. We indeed compared the current results from Jordan to previous findings from Jordan, Egypt and Kenya and Namibia which provides the discussion with the needed context and depth as recommended by the reviewer

Comment: Cultural and Behavioral Explanations: The paper mentions cultural barriers to BSE uptake but does not discuss at length specific cultural beliefs that may discourage self-examination. Further explanation on this issue would provide more insight.

Reply: We thank the reviewer for this comment regarding cultural norms as barriers to BSE practice. We agree with the reviewer that these barriers may constitute a major concern toward, not only BSE practice, but also toward research in this field. We reorganized the cultural barriers previously stated in the introduction section to follow a more thematic structure as following:

“Barriers to BSE in Jordan include cultural, structural, and informational challenges [17]. Cultural norms, such as embarrassment, patriarchal family dynamics, and the need for male approval, often restrict women’s autonomy in health-related decisions. Additionally, religious beliefs, stigma, fatalistic attitudes, and reliance on traditional medicine diminish the perceived importance of screening.”

Although we did not restate the list of cultural barriers within the discussion section to avoid redundancy within the manuscript which may negatively impact readability, we revised and edited study limitations to offer a more extensive discussion of how these cultural norms and family dynamics can potentially lead to response bias and self-selection bias:

“Moreover, cultural barriers and reluctance to discuss sensitive health issues, including breast-related issues, may have led to response bias, potentially affecting the accuracy of the results. These cultural barriers can vary between urban and rural areas, where social norms and family dynamics are often more established in the rural areas. In addition, women who get exposed to health education campaigns may be more willing to respond to such survey which indicates the possibility of self-selection bias.”

Comment: Implications for Public Health Interventions: Although recommendations are given, real-world applications to healthcare providers, policymakers, and NGOs in Jordan must be more clearly outlined

Reply: We appreciate the reviewer comment and we agree that it is necessary to outline real-world applications within the recommendations section. Therefore, we added a call for healthcare providers to educate their patients and guide them regarding BSE practices:

“We encourage healthcare providers to educate their patients on BSE practice and provide guidance for them during their interactions with the healthcare system, particularly routine check-ups and family planning visits.”

We also added a call for governmental, non-governmental and community-based organization to participate actively in tailoring targeted initiatives to serve populations that show lower BSE rates:

“Lastly, tailored interventions in regions like Irbid and Mafraq, where BSE rates are lower, could address cultural and regional differences through localized awareness campaigns. The JBCP initiatives can play a major role in tailoring such interventions; however, community-based organizations and non-governmental organizations (NGOs) can also have significant contributions to this field. Primary healthcare centers in these areas should be advised to strengthen breast cancer health education initiatives to increase awareness about screening, including BSE.”

Comment: Ethical Approval and Consent: The study used secondary JDHS data that is in the public domain. However, a more explicit explanation of ethical clearance procedures followed by the JDHS would facilitate transparency

Reply: We appreciate the reviewer recommendation to elaborate on ethical considerations, including ethical approval and consent. We revised and edited the methods section to add a new subsection concerned with the ethical considerations of the current research work, including that’s of the JPFHS report and DHS program:

“The DHS program provided ethical approval for the JPFHS before the Jordanian Department of Statistics started data collection. Informed consent was a prerequisite for an individual to participate in the survey. The current study constituted a secondary analysis of de-identified data of JPFHS 2023. The DHS program provided the data and approved the research proposal of the present work. Given the anonymized nature of data, no additional ethical approval was required to analyze it.”

Comment: Strengthen the discussion on limitations, particularly for cross-sectional design and self-reporting data.

Reply: We appreciate the reviewer comment and we understand the importance of stating limitation of the research work to avoid overstating conclusions. In addition to limitations of the cross-sectional design and self-reported data, we also added the limited data on many other variables that may be considered as confounders. Also, discussion of response bias, self-selection bias and cultural barriers was added to the limitations section:

“However, there are several limitations to consider. Being a cross-sectional study, it can only establish associations rather than causal links. Additionally, self-reported data on BSE practices may be subject to recall and social desirability biases. The study also did not account for individual-level factors, such as personal access to healthcare services and screening facilities, the quality of health education campaigns and previous contribution to health education campaigns. which could provide deeper insights into the regional disparities in BSE practices. Other factors that were not considered in the current study include family history of breast cancer, awareness of breast cancer warning signs and cultural beliefs. Moreover, cultural barriers and reluctance to discuss sensitive health issues, including breast-related issues, may have led to response bias, potentially affecting the accuracy of the results. These cultural barriers can vary between urban and rural areas, where social norms and family dynamics are often more established in the rural areas. In addition, women who get exposed to health education campaigns may be more willing to respond to such survey which indicates the possibility of self-selection bias. If women who were more familiar with the concept of BSE were the primary respondents with underrepresentation of those who show limited awareness, this may contribute to the observed non-significant association between the type of residence and BSE practice.”

Comment: Describe methodological details, e.g., how multicollinearity was accounted for in regression models.

Reply: We thank the reviewer for this comment. Multivariable logistic regression was performed using SPSS as described. Multicollinearity was not formally assessed in this analysis. This has been noted as a limitation in the Discussion section.

Comment: Discuss regional and cultural determinants of BSE practices to provide more meaningful contextual information.

Reply: We thank the reviewer for this comment and we agree that it is important to discuss regional and cultural determinants of BSE. We elaborated more on regional disparities and variations in social norms and family dynamic between rural and urban areas in the discussion section. Also, we revised and edited our limitations section to add more details regarding how these disparities may lead to potential response and self-selection biases in the studied dataset:

“Moreover, cultural barriers and reluctance to discuss sensitive health issues, including breast-related issues, may have led to response bias, potentially affecting the accuracy of the results. These cultural barriers can vary between urban and rural areas, where social norms and family dynamics are often more established in the rural areas. In addition, women who get exposed to health education campaigns may be more willing to respond to such survey which indicates the possibility of self-selection bias. If women who were more familiar with the concept of BSE were the primary respondents with under

---

## [Decision Letter · Decision Letter 1]

8 Jan 2026

PONE-D-25-00461R1

Exploring Socio-Demographic Determinants of Breast Self-Examination Practices Among Jordanian Women: Insights from the 2023 Population-Based Survey

PLOS One

Dear Dr. Shaaban Abdelgalil,

Thank you for submitting your manuscript to PLOS ONE. After careful consideration, we feel that it has merit but does not fully meet PLOS ONE’s publication criteria as it currently stands. Therefore, we invite you to submit a revised version of the manuscript that addresses the points raised during the review process.

We look forward to receiving your revised manuscript.

Kind regards,

Nülüfer Erbil, Prof.

Academic Editor

PLOS One

Journal Requirements:

Additional Editor Comments:

Dear Author,

Thank you for submitting your manuscript entitled “Exploring Socio-Demographic Determinants of Breast Self-Examination Practices Among Jordanian Women: Insights from the 2023 Population-Based Survey" to PLOS One. The manuscript has now been reviewed by expert referees, and their reports have been carefully evaluated by the editorial team.

Based on the reviewers’ comments, we are pleased to inform you that your manuscript is considered suitable for publication pending minor revisions. The reviewers’ suggestions are intended to improve the clarity, presentation, and overall quality of the paper.

We kindly ask you to revise your manuscript by addressing all reviewer comments. Please ensure that:

All reviewer points are responded to clearly and concisely.

Minor corrections and revisions are incorporated into the manuscript.

A separate response-to-reviewers document is provided, indicating how each comment has been addressed.

Please submit the revised version of your manuscript. Once received, the revision will be assessed by the editorial office, and further review may not be necessary.

We appreciate your contribution to our journal and look forward to receiving your revised manuscript.

Sincerely,

Nülüfer Erbil, Prof.

Academic Editor

Reviewer's Responses to Questions

**Comments to the Author**

1. If the authors have adequately addressed your comments raised in a previous round of review and you feel that this manuscript is now acceptable for publication, you may indicate that here to bypass the “Comments to the Author” section, enter your conflict of interest statement in the “Confidential to Editor” section, and submit your "Accept" recommendation.

Reviewer #1: All comments have been addressed

Reviewer #3: All comments have been addressed

2. Is the manuscript technically sound, and do the data support the conclusions?

Reviewer #1: Yes

Reviewer #3: Yes

3. Has the statistical analysis been performed appropriately and rigorously? 

Reviewer #1: Yes

Reviewer #3: Yes

4. Have the authors made all data underlying the findings in their manuscript fully available?

Reviewer #1: Yes

Reviewer #3: Yes

5. Is the manuscript presented in an intelligible fashion and written in standard English?

Reviewer #1: Yes

Reviewer #3: Yes

6. Review Comments to the Author

Reviewer #1: Congratulations on addressing all the comments spelled out in the manuscript. You have done a great work, and I recommend that the manuscript should be accepted for publication.

Reviewer #3: Thank you for your careful and thoughtful revisions. The revised manuscript is much improved in both clarity and coherence. Your incorporation of previous reviewer feedback is appreciated, and your study continues to offer valuable insights into predictors of Breast Self-Examination (BSE) behavior among Jordanian women.

Below are a few minor editorial corrections to further improve the manuscript prior to publication:

Terminology Consistency

Please ensure consistent use of the term “Jordan Population and Family Health Survey (JPFHS)” throughout the text. The abbreviation “DHS” is used in some places—this should be harmonized to avoid confusion.

Grammar

Original: “reading newspaper or magazines”

Suggested: “reading newspapers or magazines”

Discussion Section – Clarification

Consider rephrasing the sentence summarizing non-significant findings to improve clarity:

“Several factors, such as BMI, residence type, marital status, and pregnancy history, were not significantly associated with BSE in the final regression model.”

Table Titles

Table 1: You may consider refining the title to:

“Sociodemographic characteristics by BSE status”

Table 2:

“Multivariate logistic regression results for predictors of BSE”

Punctuation

Please ensure consistency in the use of commas in lists throughout the manuscript (e.g., “education level, internet use, and newspaper reading”).

Overall, the manuscript is strong and requires only these minor adjustments. No further revisions to content or analysis are needed.

7. PLOS authors have the option to publish the peer review history of their article (what does this mean?). If published, this will include your full peer review and any attached files.

Reviewer #1: No

Reviewer #3: No

---

## [Author Response · Author response to Decision Letter 2]

19 Mar 2026

Dear Editor in Chief:

We truly appreciate the time and effort spent by you and the reviewers. We have carefully addressed the reviewers’ comments in a point-by-point format below. We hope that our revised manuscript will be satisfactory for publication.

Best wishes.

Reviewer #1:

Comment: Congratulations on addressing all the comments spelled out in the manuscript. You have done a great work, and I recommend that the manuscript should be accepted for publication.

Reply: We thank the reviewer for the positive feedback on the revised manuscript and appreciate the positive recommendation. We truly appreciate the time you spent to provide a comprehensive assessment to the manuscript during the peer review process.

Reviewer #3:

Comment: Thank you for your careful and thoughtful revisions. The revised manuscript is much improved in both clarity and coherence. Your incorporation of previous reviewer feedback is appreciated, and your study continues to offer valuable insights into predictors of Breast Self-Examination (BSE) behavior among Jordanian women.

Reply: We thank the reviewer for the kind words and the provided comprehensive assessment to the manuscript. We have carefully addressed the comments and the recommendations in a point-by-point format below.

Comment (Terminology Consistency): Please ensure consistent use of the term “Jordan Population and Family Health Survey (JPFHS)” throughout the text. The abbreviation “DHS” is used in some places—this should be harmonized to avoid confusion.

Reply: We thank the reviewer for this comment which helped us improve consistency and avoid confusing readers. We revised the manuscript and revised Jordan Demographic and Health Survey (DHS) into Jordan Population and Family Health Survey (JPFHS) wherever it was meant to refer to the Jordanian dataset. Particularly, lines: 39, 40, 111 and 179.

Comment (Grammar):

Original: “reading newspaper or magazines”

Suggested: “reading newspapers or magazines”

Reply: We appreciate the reviewer for this comment and agree that the phrase needed grammatical correction in some parts of the manuscript. We revised the phrase to “reading newspapers or magazines” as recommended in Table 2 and lines 58 and 474.

Comment (Discussion Section – Clarification):

Consider rephrasing the sentence summarizing non-significant findings to improve clarity:

“Several factors, such as BMI, residence type, marital status, and pregnancy history, were not significantly associated with BSE in the final regression model.”

Reply: We thank the reviewer for this comment since it helped us improve the discussion of the non-significant findings. We revised the sentence to: “Several factors, such as BMI, residence type, marital status, employment status and pregnancy history, were not significantly associated with BSE in the final regression model”.

Comment (Table Titles):

Table 1: You may consider refining the title to:

“Sociodemographic characteristics by BSE status”

Table 2:

“Multivariate logistic regression results for predictors of BSE”

Reply: We thank the reviewer for this recommendation. We revised the titles to “Sociodemographic characteristics by BSE status” and “Multivariate logistic regression results for predictors of BSE” for Tables 1 and 2, respectively, as suggested by the reviewer.

Comment (Punctuation):

Please ensure consistency in the use of commas in lists throughout the manuscript (e.g., “education level, internet use, and newspaper reading”).

Reply:

We thank the reviewer for this comment. lists were revised all over the manuscript to ensure consistency in the use of commas. It was refined particularly in lines 7, 46, 240, and 409.

Comment: Overall, the manuscript is strong and requires only these minor adjustments. No further revisions to content or analysis are needed.

Reply: We thank the reviewer for these comments which are necessary for improving the rigor and clarity of our manuscript. We believe that the revised version addressed these minor adjustment, offering a more clear and readable manuscript.

---

## [Decision Letter · Decision Letter 2]

1 Apr 2026

Exploring Socio-Demographic Determinants of Breast Self-Examination Practices Among Jordanian Women: Insights from the 2023 Population-Based Survey

PONE-D-25-00461R2

Dear Dr. Shaaban Abdelgalil,

We’re pleased to inform you that your manuscript has been judged scientifically suitable for publication and will be formally accepted for publication once it meets all outstanding technical requirements.

Kind regards,

Nülüfer Erbil, Ph.D, Prof.

Academic Editor

PLOS One

Additional Editor Comments (optional):

Reviewers' comments:

Reviewer's Responses to Questions

**Comments to the Author**

1. If the authors have adequately addressed your comments raised in a previous round of review and you feel that this manuscript is now acceptable for publication, you may indicate that here to bypass the “Comments to the Author” section, enter your conflict of interest statement in the “Confidential to Editor” section, and submit your "Accept" recommendation.

Reviewer #3: All comments have been addressed

2. Is the manuscript technically sound, and do the data support the conclusions?

Reviewer #3: Yes

3. Has the statistical analysis been performed appropriately and rigorously? 

Reviewer #3: Yes

4. Have the authors made all data underlying the findings in their manuscript fully available?

Reviewer #3: Yes

5. Is the manuscript presented in an intelligible fashion and written in standard English?

Reviewer #3: Yes

6. Review Comments to the Author

Reviewer #3: The authors have satisfactorily addressed the concerns raised in the previous round of review. The revisions have strengthened the manuscript and improved its readability and coherence. I have no additional major concerns, and I recommend acceptance of the manuscript in its current form.

7. PLOS authors have the option to publish the peer review history of their article (what does this mean?). If published, this will include your full peer review and any attached files.

Reviewer #3: No

---

## [Editor Report · Acceptance letter]

PONE-D-25-00461R2

PLOS One

Dear Dr. Shaaban Abdelgalil,

I'm pleased to inform you that your manuscript has been deemed suitable for publication in PLOS One. Congratulations! Your manuscript is now being handed over to our production team.

Kind regards,

on behalf of

Dr. Nülüfer Erbil

Academic Editor

PLOS One